# Anxiety Evolution among Healthcare Workers—A Prospective Study Two Years after the Onset of the COVID-19 Pandemic Including Occupational and Psychoemotional Variables

**DOI:** 10.3390/medicina60081230

**Published:** 2024-07-29

**Authors:** Fernanda Gil-Almagro, Fernando José García-Hedrera, Cecilia Peñacoba-Puente, Francisco Javier Carmona-Monge

**Affiliations:** 1Psychology Deparment, Universidad Rey Juan Carlos, Av. de Atenas, s/n, 28922 Madrid, Spain; fgilalmagro@gmail.com (F.G.-A.); cecilia.penacoba@urjc.es (C.P.-P.); 2Nurse Intensive Care Unit, Hospital Universitario Fundación Alcorcón, Budapest, 1, 28922 Madrid, Spain; fjgarciah@gmail.com; 3Anesthesia Department, Hospital Universitario Santiago de Compostela, Rúa da Choupana, s/n, 15706 A Coruña, Spain

**Keywords:** COVID-19 pandemic, anxiety, psychological resilience, social support, self-efficacy, cognitive fusion

## Abstract

*Background and objectives:* Although previous research has found a high prevalence of anxiety during the COVID-19 pandemic among healthcare workers, longitudinal studies on post-pandemic anxiety and predictor variables have been less abundant. To examine the evolution of anxiety in healthcare workers from the beginning of the pandemic until one and a half years later, analyzing the influence of occupational and psychosocial variables, as well as their possible predictors. *Materials and Methods*: This was a prospective longitudinal design with three periods of data collection: (1) between 5 May and 21 June 2020, (2) six months after the end of the state of alarm (January–March 2021), and (3) one year after this second assessment (April–July 2022), in which generalized anxiety (GAD-7) was evaluated, as well as occupational and psycho-emotional variables (i.e., social support, self-efficacy, resilience, and cognitive fusion) in healthcare workers in direct contact with COVID-19 patients in Spain. *Results*: A high prevalence of anxiety was found, with a clear decrease over time. Associations were found between anxiety and certain sociodemographic and work variables (i.e., years of experience, *p =* 0.046; COVID-19 symptoms, *p =* 0.001; availability of PPE, *p =* 0.002; workload, *p <* 0.001; family contagion concern, *p =* 0.009). Anxiety maintained negative relationships with social support (*p <* 0.001), self-efficacy (*p <* 0.001), and resilience (*p <* 0.001) and positive associations with cognitive fusion (*p <* 0.001). Cognitive fusion seemed to be a clear predictor of anxiety. *Conclusions*: Our findings suggest that social support, self-efficacy, and resilience act as buffers for anxiety, whilst cognitive fusion was found to be a clear risk factor for anxiety. It is important to emphasize the risk role played by cognitive fusion on HCWs as a clear risk factor for stressful work events. The findings emphasize the need to implement specific interventions to promote the mental well-being of healthcare workers, particularly in crisis contexts such as the COVID-19 pandemic.

## 1. Introduction

Data recorded in the aftermath of the COVID-19 pandemic have shown that healthcare workers (HCWs) have suffered psychoemotional disturbances derived from this stressful work situation [1,2,3], added to the complicated situation in Spain resulting from the pandemic itself. In Spain, the state of alarm was decreed on 14 March 2020, leading to a subsequent phase of home confinement of the population. The end of the state of alarm was 9 May 2020. However, movements between autonomous communities were not allowed until 21 June 2020. One of the most studied consequences on HCWs following the COVID-19 pandemic has been anxiety [4,5], having found a high prevalence among HCWs in the early stages of the pandemic. Prior to the pandemic, several studies had already shown a high prevalence of anxiety among HCWs, and this was found to be due to several work-related variables, such as the fear of making mistakes in the administration of medication, the high workload, or the carrying out of care activities in highly complex units, such as intensive care units (ICU) [6,7]. During the pandemic, new studies were developed on anxiety in HCWs and its consequences on their emotional health, showing a clear increase in its prevalence derived from the fear of contagion or bedside work with infectious patients [8,9]. Specifically, most of the studies found a high prevalence of generalized anxiety, measured mostly using the Generalized Anxiety Disorder (GAD-7) instrument [10,11]. 

During the COVID-19 pandemic, anxiety was analyzed in the general population and in HCWs, showing a higher prevalence of anxiety suffered by HCWs, being especially high in the case of physicians and nurses (OR = 1.9, 95% CI: 1.367–2.491, *p <* 0.001) [12]. Most of the research published to date was carried out at the beginning of the pandemic, covering the period up to the summer of 2020. The data collected worldwide indicated moderate to severe anxiety, with percentages ranging between 20% and 60% [13,14,15,16,17,18,19]. In HCWs, these anxiety levels remained stable until the end of 2020 [20,21,22,23]. 

China has been one of the most studied contexts, likely because it is the country where the pandemic started, presenting high anxiety averages for HCWs. In China, a prevalence of anxiety between 45% and 60% was found in HCWs in 2021 [24], which is considerably higher than that of the general population of China [25]. Anxiety studies conducted in China subsequently, coinciding with new hospitalization peaks, reflect figures between November 2022 and February 2023 of close to 50% anxiety prevalence among HCWs [26].

However, the study of long-term anxiety after the critical stage of the pandemic has not been frequent. To understand the real impact and the variables involved in the evolution of anxiety, studies based on the time after the acute stress of the onset of the pandemic, which could reveal factors that act as buffers or chronifiers of anxiety, are essential. Thus, longitudinal studies based on the long-term evolution of anxiety and the variables involved are necessary, as the only scientific evidence currently available is focused on the evolution of pandemic anxiety until 2021 [27,28].

Different longitudinal studies have shown a prevalence of anxiety in nurses of 43.1% during confinement and 16% a few months later [29]. Research carried out in the USA also found a decrease in the levels of generalized anxiety, from 46.3% to 23.2% [28]. Longitudinal studies carried out in Spain suggest that the improvement in anxiety levels over time is not so evident, indicating similar percentages of anxiety prevalence during the first and second waves [10].

Studies carried out in Spain and the U.S.A. that analyzed the influence of sociodemographic and occupational variables on the anxiety suffered by HCWs during the COVID-19 pandemic have suggested that several variables acted as risk factors, some of which were the fear of contagion, the development of care activities with infectious patients, the high workload, or the scarcity of personal protective equipment (PPE) [30,31]. Other studies from China have found that working in highly complex units such as the ICU is an added risk factor for the development of anxiety [32]. Similarly, a systematic review including different studies from China and Singapore indicated a higher prevalence of anxiety in nurses than in physicians during this time [9]. 

However, studies analyzing the influence that different psychosocial or personality variables have had on anxiety during the COVID-19 pandemic are not as abundant. Most of the available studies have focused on the influence of resilience, stating that low levels of resilience negatively affected the evolution of anxiety in HCWs at this time [33]. Likewise, different studies have pointed out the positive effect that social support, hobbies, or a healthy lifestyle had on the evolution of anxiety [34]. The study of cognitive fusion has shown a clear relationship between high levels of cognitive fusion and high levels of anxiety among HCWs [35,36]. Meanwhile, self-efficacy allows cushioning of the negative consequences of anxiety, such as burnout, facilitating better management of negative emotional states [37]. 

Most studies evaluating the influence of different psychosocial or personality variables on anxiety have focused on a specific time point, without considering the consequences that these variables may have had on the evolution of anxiety in later phases. Some studies call these later phases of the pandemic “the post-pandemic era of COVID-19” [38]. Because of this approach, it is particularly relevant to conduct prospective studies in the mid- and long-term, allowing us to explore not only the evolution of anxiety and its possible chronification but also the possible precipitating or dampening factors involved in its evolution.

The aim of the present study was to analyze the evolution of anxiety, using the GAD-7 questionnaire, in a sample of HCWs who were in direct contact with infectious patients during the first stage of the COVID-19 pandemic. The professionals were followed up at two additional time points, six months after the initial assessment and one year later. As additional aims, we analyzed the association of anxiety and its evolution with sociodemographic, occupational, and personality variables in order to find out the possible predictors that affected the evolution of anxiety among HCWs during and after the COVID-19 pandemic.

## 2. Materials and Methods

### 2.1. Design

A prospective longitudinal study was carried out with three data collection periods: (1) between 5 May and 21 June 2020 (the final phase of the state of alarm and confinement, declared in Spain on 14 March), (2) six months after the end of the state of alarm (January–March 2021), and (3) one year after this second assessment (April–July 2022). During the first data collection period, Spain was in a state of alarm from 14 March, which decreed a confinement phase, until 28 April, at which time a de-escalation began, culminating in the end of the state of alarm and population confinement on 21 June 2020. During the second data collection period, the situation was still complicated with 3,347,512 confirmed cases and 76,328 deaths as of 9 April 2021. The third data collection period ended with 12,973,615 confirmed COVID-19 cases and 108,730 deaths. During the three time periods, participants’ generalized anxiety was assessed for its evolution. Sociodemographic and occupational variables were evaluated at the first time point, while variables related to personality were assessed during the first and second time points (Table 1).

### 2.2. Procedure and Participants

The research team designed an electronic questionnaire for data collection in which all the variables to be studied were included and the different validated instruments were attached. Informed consent was requested from the participants, as well as an e-mail if they were interested in participating in the following phases of the research evaluation.

The sample consisted of HCWs belonging to the Spanish National Health System. Probabilistic convenience sampling was carried out with the following inclusion criteria: being a nurse, physician, or nursing care technician; carrying out healthcare activities in a public or private service of the National Health System; being 18 years old or older; and having been in direct contact with COVID-19 patients. The following criteria were used as exclusion criteria: having been on sick leave during the data collection period or performing healthcare activity in the field of health management.

A minimum study population of 120 was taken as the reference figure established for prospective studies [39]. In addition to the complicated circumstances of the COVID-19 pandemic, we have to take into account the fact that the study was conducted in a longitudinal manner [40,41]. Therefore, a minimum sample size of 400 participants was established for the first time point, obtaining a total sample of 1374 HCWs during this period. Of these, 881 continued to participate in the second time point; of these, 257 continued to participate in the third assessment, constituting the final sample of the study, well above the 120 initially estimated.

To obtain the sample, the link with the questionnaire was sent to HCWs belonging to the Spanish health system, both public and private, distributing the questionnaire through social networks (Facebook, LinkedIn, Twitter, and WhatsApp), in addition to the corporate e-mails of the different public and private services of the National Health System. For the circulation of the questionnaire during the second and third time points, the e-mails of the HCWs who had participated in the first evaluation were used, requesting their participation again in the following phases of the study.

### 2.3. Variables and Instruments

#### 2.3.1. Generalized Anxiety [Time Point 1, 2 and 3]

The presence of symptoms of generalized anxiety disorder was evaluated using the Generalized Anxiety Disorder (GAD-7) [42] in its Spanish version [43]. It consists of a 7-item scale with a Likert-type response format, consisting of a 4-point scale ranging from 0 (not at all) to 3 (almost every day), with a total score range from 0 to 21. Four severity groups are established with the following cut-off points [42]: no anxiety/minimal (0–4), mild (5–9), moderate (10–14), or severe anxiety (15–21). Internal consistency in our sample was excellent at all three time points, with Cronbach’s alpha coefficients of 0.93, 0.93, and 0.94, respectively.

#### 2.3.2. Sociodemographic and Occupational Variables [Time Point 1]

An ad hoc questionnaire developed by the research team was used to collect these data. Specifically, these data were sociodemographic data (age, gender, and family situation), work data (category, service, work experience in years, availability of PPE, workload (less, equal, or greater than usual)), and concerns about contagion (their own or a family member’s (with a 4-point Likert-type response format (from 1 “not at all concerned” to 4 “very concerned”)).

#### 2.3.3. Personality Variables [Time Points 1 and 2]

-Social support [time point 1]: measured using the Spanish version [44] of the Multidimensional Scale of Perceived Social Support (MSPSS) [45], which is composed of 12 items divided into three dimensions: family, friends, and significant others, with a 7-point Likert-type response scale (from 1 “completely disagree” to 7 “completely agree”). The final score comes from the sum of its three subscales. The instrument has good properties [46,47], and for our study, its reliability was α = 0.85 for the general questionnaire, while for the subscales, the α values obtained were 0.81, for family, 0.82 for friends, and 0.79 for significant others.-Self-efficacy [time point 1]: The Spanish version of the General Self-Efficacy Scale (GSES) [48] was used, consisting of 10 Likert-type items scoring from 1 “completely disagree” to 4 “completely agree,” with the total score ranging from 10 to 40. This instrument, in our study, presented high internal consistency α = 0.91.-Resilience [time point 1]: we used the Spanish version of the Resilience Questionnaire (RS-14) [49], made up of 14 Likert-type items with 7 alternatives, scoring from 1 “strongly disagree” to 7 “strongly agree”, with a total score ranging from 14 to 98, whereby higher scores indicate greater resilience. In our study, α was 0.94.-Cognitive Fusion [time point 2]: The Spanish version [50] of the Cognitive Fusion Questionnaire (CFQ) [51] was administered, which is made up of 7 Likert-type items with 7 response options, ranging from 1 “never” to 7 “always”, whereby higher scale scores imply a higher degree of cognitive fusion. A Cronbach’s α of 0.97 was obtained for our study.

### 2.4. Data Analysis

Descriptive analysis and Cronbach’s alpha were performed. Qualitative variables were described with frequencies (n) and percentages (%) and quantitative variables with means (M) and standard deviations (SD). To analyze the bivariate association between variables (analysis of possible covariates), Student’s *t*-test, one-factor analysis of variance (ANOVA), and Pearson’s correlations were used, depending on the nature of the variables analyzed. A linear regression analysis was performed to define the weight of the personality variables at each of the time points, following the stepwise method to introduce the predictor variables. Statistical analysis was performed with the Statistical Package for the Social Sciences (SPSS), version 21 for Windows. The results were considered statistically significant for values of *p* < 0.05.

## 3. Results

### 3.1. Description of the Sociodemographic, Occupational, and Personality Variables of the Sample

Table 2 shows the sociodemographic, occupational, and health data of the 257 participating HCWs, represented by frequencies, percentages, means, and SD. Of them, 210 (81.7%) were female and the mean age was 43.67 years old (SD 9.78). Most participants were nurses 151 (58.8%), followed by physicians 65 (25.3%), whilst 41 (16.0%) were other types of HCWs. The most represented service was the ICU with 94 HCWs (36.6%), followed by hospitalization with 73 HCWs (28.4%). The mean number of years of experience in the service in which they worked was 10.70 (SD 9.23). Of the sample, 195 HCWs (75.9%) were very worried about their own and/or a family member’s infection. Fifty-one professionals (19.8%) requested psychological help. The scores of the instruments used to describe the different personality variables of the participants are shown in Table 2.

### 3.2. Description of Anxiety in HCWs and Its Evolution over Time

The sample presented the highest mean score for the anxiety scale (10.82; SD = 5.86) at the first time point, and for the score compatible with severe anxiety symptoms, a downward trajectory was found at the following data collection points (Table 3 and Figure 1). At the third time point (T3), HCWs presented a mean score of 7.61 (SD = 5.13), compatible with moderate anxiety symptoms. Statistically significant differences were observed between the three time points. Table 4 shows the evolution over time of anxiety levels. Moderate and severe anxiety were more prevalent at T1, whilst mild and moderate anxiety were more frequent at T3. Thus, at T1, the total number of HCWs with symptoms compatible with moderate and severe anxiety was 148 (57.6%) and with minimal and mild anxiety was 109 (42.4%), while these values appeared inverted at T3, reaching 32.7% and 67.3%, respectively.

### 3.3. Associations between Anxiety and Sociodemographic, Professional, Occupational, Health, and Personality Variables of the Sample

Table 1 shows the relationship between anxiety at the three time points and sociodemographic, occupational, and psychosocial variables. Women showed significantly higher anxiety scores than men at all three time points (*p* < 0.004). Physicians showed lower anxiety scores than the other HCWs at T1 (*p* = 0.048), a difference that does not occur at any other time point with any other HCW. At none of the time points were there differences in anxiety scores depending on the service in which HCWs performed their activity.

Although it does not hold at all time points, work experience in the current unit at T1 was significantly associated with anxiety (*p* = 0.046), whereby professionals with less experience had higher anxiety scores. Higher workload was significantly related to higher anxiety scores at T1 and T2 (*p* < 0.001). Similarly, the lack of availability of PPE was significantly related to higher anxiety scores at all time points (*p* = 0.002).

HCWs who sought psychological help at all time points showed significantly higher anxiety scores (*p* = 0.001). Social support was associated with lower anxiety scores, mainly on the friend subscale, which was significant at all time points (*p* < 0.001).

In relation to the psychological variables, all of them were significantly associated with anxiety at all time points. Self-efficacy and resilience presented a significant and negative correlation (r = −0.347, *p* < 0.001; r = −0.269, *p* < 0.001) while cognitive fusion presented a significant and positive correlation (r = 0.539, *p* < 0.001).

### 3.4. Linear Regression Analysis between Anxiety and Personality Variables

A linear regression using a stepwise approach was carried out for anxiety and the different psychological variables that presented significant associations with it at each of the time points. The final models are presented in Table 5, including only the variables that were statistically significant in the proposed models. The model explained 35.2% of the variance at T1, 51.1% of the variance at T2, and 27.2% of the variance at T3. 

## 4. Discussion

In the present study, a group of HCWs working with COVID-19 patients at the beginning of the pandemic were followed up over time (more than two years) to evaluate the evolution of their anxiety, as well as to assess possible factors that may help to control or worsen it. In general, our results show a decrease in the levels of anxiety perceived by HCWs, although with a smaller reduction than that found in other studies carried out with a shorter follow-up time [31,52]. A relevant aspect of our study was the inclusion of cognitive fusion, which has only very recently been studied in the literature. In our sample of HCWs, this variable was shown to be a precipitator of anxiety, interfering in its evolution, having found that HCWs with high levels of cognitive fusion presented worse anxiety evolution. In addition, self-efficacy, resilience, and social support from friends were shown to be buffers.

As already mentioned, anxiety is a very common symptom among HCWs, the prevalence of which increased during the COVID-19 pandemic due to several socio-occupational factors [30,53]. However, the number of studies that have attempted to carry out a long-term follow-up of this situation has been very scarce, making it difficult to define psychological, occupational, or personal aspects that may facilitate or protect the appearance of this type of disorder, with this being one of the main strengths of our study.

The present study shows a significant decrease in the anxiety levels of HCWs across the three time points, showing lower means for anxiety symptoms in the last time point compared to the beginning of the pandemic, with statistically significant differences between each time point. These results are consistent with previous research assessing the evolution of anxiety in HCWs [28].

With regards to the possible sociodemographic and occupational variables involved, our findings show the association of gender; specifically, being a woman was predisposed to developing increased levels of anxiety. Previous studies conducted in Taiwan support a clear association between being female and higher levels of anxiety [6]. In our study, younger professionals had higher levels of anxiety at the first time point. In terms of the professional category, nurses and nursing care technicians had higher anxiety scores at the three time points compared to physicians, with this difference being significant at the first time point (*p* = 0.048). Previous research has also placed the youngest professionals at the top of the list [54,55] and bedside caregivers as the category most likely to experience elevated levels of anxiety following a stressful work event such as the COVID-19 pandemic [9]. 

As far as service is concerned, our findings suggest that anxiety levels do not seem to be related to the unit in which the care services are performed (considering, in this case, intensive care units (ICUs), hospitalization, emergency, and primary care). This result seems contradictory to some research that has shown an association between the nature of the work environment and anxiety in nurses, stating, specifically, that nurses working in the ICU reported higher levels of anxiety compared to those working in other hospital services [56,57]. Significant associations have also been found between working in critical care units and high levels of anxiety in nurses [7]. Other studies have indicated that the stress inherent in emergency settings may contribute to higher levels of anxiety in nurses working in these services [58]. These results are not in accordance with our findings and suggest that the association between work environment and anxiety among HCWs may be more complex than previously considered, varying from one specific context to another and depending on the individual characteristics of the HCWs. Furthermore, we also believe that the time point at which these assessments are made should also be considered. Regarding the sociodemographic and occupational variables assessed, our findings show that the work overload experienced by HCWs throughout the COVID-19 pandemic had a significant direct relationship with anxiety throughout the three time points, which is in line with previous research associating perceptions of high workload with high levels of anxiety [59].

Our results also point to additional variables that are particularly relevant to anxiety experienced at the first time point, such as work experience, which had a significant negative relationship with anxiety. This significant relationship disappeared in the second time point, likely linked to the learning and development of adaptive strategies to cope with anxiety. Our results support the trend observed in previous research that similarly shows a relationship between less work experience and higher levels of anxiety in nurses [30]. These findings suggest that anxiety may be more prevalent in inexperienced HCWs, perhaps due to a lack of adaptation to the work environment or the complexity of specific units [60]. The relationship between the unavailability of PPE and concerns about contagion of family members with regard to anxiety was significant and positive throughout the study, as previous studies have shown [61,62].

Regarding the role of psychosocial variables, bivariate analyses indicated that social support behaved as a protective variable for anxiety, although it is necessary to take into account its multidimensional nature. In this case, the social support of friends was particularly relevant, given that it maintained significantly negative relationships with anxiety at the three time points [63,64]. Thus, considering this multidimensional nature, in the first time point, social support played a protective role for anxiety in all its spheres, with larger effect sizes with regards to total social support and the social support of friends. During the pandemic, the role of social support in HCWs was widely studied, with studies finding it played a protective role against psychoemotional alterations derived from work stress, as is the case of anxiety [65,66]. Within our results, it is interesting to observe how this relationship between social support and anxiety disappears over time, with only social support from friends maintaining an inversely significant relationship with anxiety throughout the study. Different authors point out the importance of the social support derived from friends in transit through stressful situations, defining it as a clear buffer of anxiety [63]. 

Regarding the effect of self-efficacy on anxiety, the results of the univariate analyses indicated that it behaves as a protective trait over time and that it acts as a clear buffer against anxiety for HCWs in situations of high occupational stress. However, in the regression models, this effect was not maintained, so the protective effect initially identified may have been derived from other interactions or variables with greater weight in the final models. However, previous research has analyzed the role of self-efficacy in nurses, corroborating the protective role played by self-efficacy on HCWs, not only in the reduction in emotional symptoms but also in the development of strength in the face of stressful work situations [37]. 

Resilience has also been a well-studied trait of HCWs throughout the pandemic [61,67]. Our results from univariate analyses revealed that resilience was negatively related to anxiety at all three time points, as nurses who showed higher levels of resilience maintained lower anxiety scores. These results are consistent with previous research conducted throughout the pandemic on HCWs, which reflects the relevance of training HCWs in resilience for more adaptive coping and less distress in stressful work situations [61,67]. However, when performing regression models, this effect did not hold and did not appear to have an effect on the evolution of anxiety over time in our HCWs sample.

Within our study, the influence of cognitive fusion on the evolution of anxiety was assessed as a personality trait. Our results suggest that cognitive fusion is a clear precipitator of anxiety, maintaining significant positive relationships with anxiety throughout the three time points. Although cognitive fusion is a trait that is not well recorded in the existing literature, some research already points to the clear positive association between cognitive fusion and anxiety [68]. Studies on HCWs during the pandemic have found a negative effect of thought rumination on anxiety [36,69]. Only through the longitudinal nature of our study can it be affirmed that cognitive fusion represents a clear risk factor for the development of anxiety derived from a stressful work event. 

Finally, it is necessary to point out some of the limitations of our research. Among them, we can highlight the non-probabilistic convenience sampling, which limits the generalization of the results. In addition, the low participation of males may lead to a bias in terms of gender analysis, although this low representation corresponds to the reality of the profession. On the other hand, it would have been of interest to obtain previous (baseline) assessments of anxiety of the professionals who participated in the study from before the pandemic. The loss of participants over the course of the study could also have been a source of bias.

## 5. Conclusions

In contrast to the abundance of cross-sectional studies documenting anxiety in HCWs, there is a notable paucity of longitudinal research examining its evolution over time. These studies are essential to understanding the dynamics of anxiety, identifying risk and protective factors, and developing effective interventions [70]. Without this longitudinal understanding, it is difficult to determine whether current interventions are effective or whether new approaches are needed to adequately address anxiety in HCWs. The present study, in addition to identifying occupational risk factors documented in previous research, points to the protective role of resilience, self-efficacy, and especially, social support (from friends), in addition to marking a clear negative predictor in the evolution of anxiety such as cognitive fusion.

Anxiety in HCWs not only affects their own well-being but can also have negative consequences on the quality of care they provide to patients. The fatigue, exhaustion, and lack of focus associated with anxiety can influence clinical decision making and the ability to provide safe and effective care [71]. Therefore, interventions aimed at mitigating the anxiety of HCWs are important not only for their own health but also for the general quality of medical care. 

## Figures and Tables

**Figure 1 medicina-60-01230-f001:**
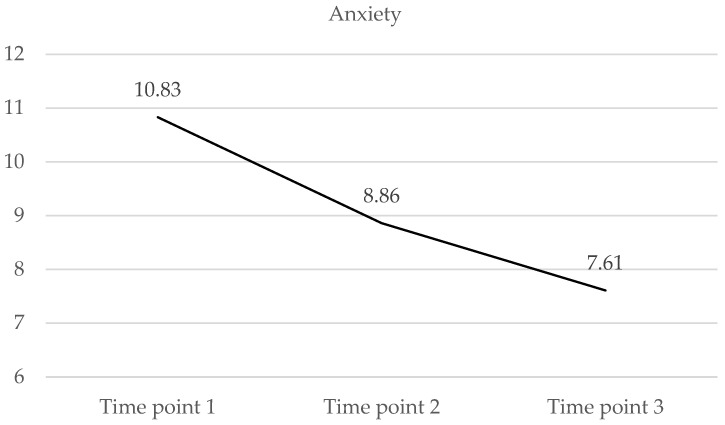
Anxiety averages for the sample at the different time points.

**Table 1 medicina-60-01230-t001:** Variables collected at different time points.

	Variables Collected at Different Time Point		
	1st Evaluation Period	2nd Evaluation Period	3rd Evaluation Period
	5 May–21 June (2020)	9 January–9 April (2021)	11 April–15 July (2022)
Symptoms	Anxiety	Anxiety	Anxiety
Sociodemographics	Age, gender, family situation, work experience, job category, service, workload, avaliability PPE, concern about contagion, request of psychological support		
Personality	Resilience Self-efficacy Social Support		Cognitive Fusion

**Table 2 medicina-60-01230-t002:** Sociodemographic and health characteristics. Association between the different variables and anxiety.

				Anxiety									
				Time Moment 1				Time Moment 2			Time Moment 3		
		f (%)	Mean (SD)	Mean (SD)	Test		*p*	Mean (SD)	Test	*p*	Mean (SD)	r^2^	*p*
Age			43.68 (9.78)		r^2^	−0.132	0.034		−0.089	0.153		−0.018	0.773
Experience (years)			10.70 (9.23)		r^2^	−0.125	0.046		−0.033	0.600		−0.031	0.626
Gender	Man	47 (18.3%)		7.95 (5.84)	t	−3.746	<0.001	6.04 (4.91)	−3.965	<0.001	5.54 (5.10)	−2.993	0.003
	Woman	210 (81.7%)		11.46 (5.68)				9.47 (5.33)			8.06 (5.04)		
Professional Category	Physician	65 (25.3%)		9.29 (5.44)	F	3.075	0.048	8.32 (5.06)	0.503	0.605	6.78 (4.89)	1.238	0.292
	Nurse	151 (58.8%)		11.27 (5.89)				9.12(5.49)			7.80 (4.90)		
	Nursing tecnician	41 (16.0%)		11.61 (6.10)				8.73 (5.72)			8.22 (6.20)		
Cohabitation	Without a partner	77 (30.0%)		9.97 (5.87)	t	−1.526	0.128	8.321 (5.37)	−1.030	0.304	6.87 (5.14)	−1.518	0.130
	With a partner	180 (70.0%)		11.19 (5.84)				9.08 (5.42)			7.93 (5.11)		
Workload	Lower than usual	19 (7.4%)		6.21 (5.39)	t	−5.489	<0.001	5.05 (4.12)	−3.366	0.001	5.63 (5.42)	−1.841	0.067
	Equal than usual	24 (9.3%)		6.87 (5.35)				7.41 (4.88)			6.83 (3.67)		
	Higher than usual	214 (83.3%)		11.68 (5.60)				9.36 (5.42)			7.87 (5.22)		
Speciality	ICU	94 (36.6%)		11.29 (5.72)	F	0.462	0.764	8.65 (4.96)	2.397	0.051	7.26 (5.51)	0.948	0.437
	Hospitalisation	73 (28.4%)		10.53 (5.94)				8.86 (5.66)			7.89 (5.06)		
	Emergencies	38 (14.8%)		9.27 (6.32)				7.50 (5.67)			6.82 (4.63)		
	Primary Care	42 (16.3%)		11.21 (5.71)				10.90 (5.25)			8.76 (4.84)		
	Others	10 (3.9%)		10.20 (6.09)				7.30 (5.74)			7.10 (5.02)		
PPE availability	Yes	107 (41.2%)		9.46 (5.54)	t	3.122	0.002	8.01 (4.76)	2.056	0.041	6.77 (4.74)	2.094	0.037
	No	150 (58.8%)		11.77 (5.91)				9.42 (5.78)			8.17 (5.34)		
Worry	Yes	195 (75.9%)		11.81 (5.76)	t	−4.968	<0.001	9.69 (5.21)	−4.563	<0.001	8.11 (5.11)	−2.819	0.005
Psychological help	Yes	51 (19.8%)		13.08 (4.71)	t	−3.601	0.001	12.63 (5.35)	−5.921	<0.001	9.14 (5.03)	−2.395	0.017
Social support	Total			5.78 (1.21)	r^2^	−0.227	<0.001		−0.098	0.117		−0.151	0.016
	Family			5.88 (1.20)	r^2^	−0.181	0.004		−0.114	0.068		−0.133	0.033
	Friends			5.64 (1.40)	r^2^	−0.325	<0.001		−0.177	0.004		−0.269	<0.001
	Significant Others			5.81 (1.55)	r^2^	−0.097	0.119		−0.019	0.763		−0.006	0.920
Resilience				78.39 (14.29)	r^2^	−0.269	<0.001		−0.242	<0.001		−0.230	<0.001
Self-Efficacy				29.18 (4.09)	r^2^	−0.347	<0.001		−0.315	<0.001		−0.318	<0.001
Cognitive Fusion					r^2^	0.539	<0.001	21.97 (10.78)	0.715	<0.001		0.431	<0.001

**Table 3 medicina-60-01230-t003:** Anxiety at each of the time points of data collection.

				Student’s *t* Test for Paired Samples					
	Time 1	Time 2	Time 3	Time 1–2		Time 1–3		Time 2–3	
	M (SD)	M (SD)	M (SD)	t	*p*	t	*p*	t	*p*
Anxiety	10.82 (5.86)	8.86 (5.41)	7.61 (5.13)	6.694	<0.001	10.377	<0.001	3.879	<0.001

**Table 4 medicina-60-01230-t004:** Anxiety level percentages at each of the time points of data collection.

		Time 1		Time 2		Time 3	
		n (%)	M (SD)	n (%)	M (SD)	n (%)	M (SD)
Anxiety			10.82 (5.86)		8.86 (5.41)		7.61 (5.13)
Grouped Anxiety	No anxi/Min ^2^	41 (16.0)		57 (22.2)		74 (28.8)	
	Mild	68 (26.5)		94 (36.6)		99 (38.5)	
	Moderate	78 (30.4)		65 (25.3)		64 (24.9)	
	Severe	70 (27.2)		41 (16.0)		20 (7.80)	
Anxiety Mode/Severe ^1^	Yes	148 (57.6)		106 (41.2)		84 (32.7)	

^1^ Anxiety Moderate/Severe ^2^ No anxiety/Minimal.

**Table 5 medicina-60-01230-t005:** Linear regression analysis between anxiety and the different personality variables.

Anxiety	*F*	R^2^	IncR^2^	Beta	*t*	*p*
Anxiety T1	45,906	0.352	3.345			
Cognitive Fusion				0.447	8.119	<0.001
–Social support friends				−0.200	−3.811	<0.001
Self-efficacy				−0.134	−2.416	0.016
Anxiety T2	266,350	0.511	0.509			
Cognitive Fusion				0.715	16.320	<0.001
Anxiety T3	23,504	0.272	0.260			
Cognitive Fusion				0.307	5.188	<0.001
Social support friends				−0.299	−4.333	<0.001
Social support significant others				0.231	3.371	0.001
Self- efficacy				−0.189	−3.157	0.002

## Data Availability

Research data will be available upon request to the corresponding author.

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
