# Peer review of "Anxiety Evolution among Healthcare Workers—A Prospective Study Two Years after the Onset of the COVID-19 Pandemic Including Occupational and Psychoemotional Variables"

_medicina, 2024, doi:10.3390/medicina60081230_

Round 1

Reviewer 1 Report

Comments and Suggestions for Authors

The paper undertakes a topic concerning evolution and factors influencing anxiety (measured by Generalized Anxiety Disorder GAD-7 scale) among healthcare workers who had contacts with ill patients during the COVID-19 pandemic. The paper considers two types of predictors, i.e. sociodemographic and occupational variables and, as named by the authors, personality variables. The latter are derived from several Likert-type scales described in the literature. The reliability evaluated by Cronbach’s alpha is at a satisfactory level. The Authors focus on estimating the association between anxiety prevalence and enumerated factors in three time points using various statistical methods (t-tests, ANOVA, correlation analysis, regression analysis).  Elements of the research worth emphasizing are: a prospective longitudinal study, analysis of the evolution of anxiety, usage of various factors as resilience, self-efficacy, social support and cognitive fusion.

I have some specific comments:

Table 1: „request of psicologichal” – there is misspelling in this phrase.

Lines 204-206: “A binary logistic regression analysis was also performed, following the Hosmer-Lemeshow goodness-of-fit method, taking anxiety as dichotomous and studying as predictor variables those that were significant in the bivariate analysis at each of the time points.” No results of a logistic regression are shown in the paper, recalled or discussed, so what is the purpose of mentioning it in the Data analysis section?

Line 213: “Table 2 shows the sociodemographic, occupational and health data” The table does not show particular data but statistical measures like mean, sd or frequencies. It would be better to precise it.

Line 223: “Table 2. Association between the different variables and anxiety” The title of the table does not reflect the whole content of the table. Respondents characteristics – descriptive statistics are also included and you write about it in lines 213-220.

Line 235: “Table 3 shows the evolution over time of anxiety levels”. Table 4, I suppose.

Line 284: “Three models were obtained at T1, one model at T2 and four models at T3.” I have serious doubts if this is correct. The results in table suggest that at T1 you created a multivariate regression model with 3 independent variables, at T2 a model with 1 independent variable and at T3 a model with 4 independent variables. I conclude so, because:

1.     In the table you provide F and R2 – one for each time point, so it suggests one model was created for each time point.

2.     In Data analysis section you provide following information (lines 202-204) “A linear regression analysis was performed to define the weight of the  personality variables at each of the time points, following the stepwise method to introduce the predictor variables.” Stepwise method is usually used to introduce several independent variables in the model step by step and finally a multivariate model is obtained.

3.      In lines 286-287 “The best model for T3, including cognitive fusion, social support (from friends and others) and self-efficacy as predictor variables, explained 27.2% of the variance” – this describes that there is one model for T3 with 3 independent variables.

Table 5 - independent variables for Anxiety T1 model are given in Spanish.

Lines 308-310: Giving a fragment of text in bold here is in my opinion unnecessary.

Line 346: Is “mare” an appropriate word here?

Lines 411- 414: This is a fragment of guidelines for authors…

Author Response

Good morning,
We appreciate the effort you have taken to review our manuscript and all the suggestions you have provided. We have incorporated all the comments suggested. We hope the new changes meet their expectations, and we hope that they consider the work apt for publication in Medicina.

Comment 1: Table 1: „request of psicologichal” – there is misspelling in this phrase.

Response: The word “psicologichal” has been spelled correctly. Thank you for your appreciation.

Comment 2: Lines 204-206: “A binary logistic regression analysis was also performed, following the Hosmer-Lemeshow goodness-of-fit method, taking anxiety as dichotomous and studying as predictor variables those that were significant in the bivariate analysis at each of the time points.” No results of a logistic regression are shown in the paper, recalled or discussed, so what is the purpose of mentioning it in the Data analysis section?

Response: Thank you for your comment. Indeed, the article does not show logistic regression results, so we have modified the methodology section to make it more consistent with the results.

Comment 3: Line 213: “Table 2 shows the sociodemographic, occupational and health data” The table does not show particular data but statistical measures like mean, sd or frequencies. It would be better to precise it.

Response: Thank you for your suggestion. We have specified that table two shows sociodemographic, occupational and health data represented through frequencies and percentages, means and SD.

Comment 4: Line 223: “Table 2. Association between the different variables and anxiety” The title of the table does not reflect the whole content of the table. Respondents characteristics – descriptive statistics are also included and you write about it in lines 213-220.

Response: Thank you for your appreciation. We have modified the title of table two to make it more coherent.

Comment 5: Line 235: “Table 3 shows the evolution over time of anxiety levels”. Table 4, I suppose

Response: Thank you for your comment. It is indeed table 4. The error has been corrected in the manuscript. Thank you very much.

Comment 6: Line 284: “Three models were obtained at T1, one model at T2 and four models at T3.” I have serious doubts if this is correct. The results in table suggest that at T1 you created a multivariate regression model with 3 independent variables, at T2 a model with 1 independent variable and at T3 a model with 4 independent variables. I conclude so, because:

  1. In the table you provide F and R2 – one for each time point, so it suggests one model was created for each time point.
  2. In Data analysis section you provide following information (lines 202-204) “A linear regression analysis was performed to define the weight of the  personality variables at each of the time points, following the stepwise method to introduce the predictor variables.” Stepwise method is usually used to introduce several independent variables in the model step by step and finally a multivariate model is obtained.
  3. In lines 286-287 “The best model for T3, including cognitive fusion, social support (from friends and others) and self-efficacy as predictor variables, explained 27.2% of the variance” – this describes that there is one model for T3 with 3 independent variables.

Response:

The section has been fully edited as the information included was not correct. We performed one linear regression analysis for each of the time points (T1, T2 and T3) and we included all the variables that had significant association with anxiety using a stepway analysis. Only the variables that were significant in the model were included in table 5.

Comment 7: Table 5 - independent variables for Anxiety T1 model are given in Spanish.

Response: Thank you for your appreciation. The independent variables for anxiety at T1 have been translated into English.

Comment 8: Lines 308-310: Giving a fragment of text in bold here is in my opinion unnecessary

Response: Thank you for your comment. The bold font has been removed.

Comment 9: Line 346: Is “mare” an appropriate word here?

Response: Thank you for your appreciation. The word “mare” has been replaced by “made”.

Comment 10: Lines 411- 414: This is a fragment of guidelines for authors…

Response: Thank you very much for your comment. The fragment of guidelines for authors has been removed from the manuscript.

Reviewer 2 Report

Comments and Suggestions for Authors

Thank you for the opportunity to review this manuscript. 

There are just a few comments: 

1. The use of n and % needs to be consistent in the results section particularly in 3.1 the first paragraph. 

2. Ethics: There needs to be mention of how ethical consent was obtained and who granted permission for  this study to proceed. 

Otherwise this is an interesting article. Well done.

Author Response

We appreciate the effort you have taken to review our manuscript and all the suggestions you have provided. We have incorporated all the comments suggested. We hope the new changes meet their expectations, and we hope that they consider the work apt for publication in Medicina.

Comment 1: The use of n and % needs to be consistent in the results section particularly in 3.1 the first paragraph

Response: Thank you very much for your comment. We have modified the text to make it more consistent.

Comment 2: Ethics: There needs to be mention of how ethical consent was obtained and who granted permission for this study to proceed. 

Response: Thank you for your appreciation. The section of the manuscript entitled “Institutional Review Board Statement” includes all the information on the two Ethics Committees that reviewed the present study, as well as dates and reference codes.

“This study was approved by the Ethics Committee (Ref: 20/88) and ratified by the Central Re-search Commission (Ref: 28/20) in order to disseminate the questionnaire to primary care nursing professionals. At the beginning of the questionnaire, all participants were informed of the objec-tive and procedure of the research, and their consent was requested, as well as the possibility of contacting them again by e-mail, given the longitudinal nature of the study. The study was sup-ported by the Spanish Society of Intensive Care Nursing and Coronary Units (SEEIUC), which collaborated with the dissemination of the study.

The present study followed national and international deontological guidelines, the Helsinki dec-laration and the Code of Good Practice and Order SAS/3470/2009. The processing of the personal data of the study participants complied with Organic Law 15/1999, of 13 December, on the Pro-tection of Personal Data (LOPD) and with Regulation no. 2016/679 of the European Parliament and of the Council, of 27 April, on Data Protection (GDPR).

Ethical approval board name: Hospital Universitario Fundación Alcorcón, Code 20/88, Date: 1 May 2020.

Reviewer 3 Report

Comments and Suggestions for Authors

The test is interesting. However, the whole article needs to be more precise.

1.The abstract states that the study was conducted from the beginning of the pandemic. Please write clearly what time period you are referring to, the beginning and the end. The country in which the study was conducted is not stated, and the conclusions in the abstract are general in nature.

 2.In the introduction it would have been appropriate to write from two sentences about the different periods of pandemic restrictions and restrictions.

3.In line 54 is given the date 2020, and in 59 the date 21. Does the text correspond well with each other?

4.Line 63 is not precise.

5.In line 76 to 82 there are no specific countries, and in the earlier paragraphs the names are given. It would be worthwhile to harmonize the text.

6. In the 94th line we read about many countries, and we have one footnote 39. So what does that mean many countries?

7. How to explain line 130, where the questionnaire was created ad hoc?

8. Item 4 is titled Discussion. Shouldn't there be added restrictions that are written about at the end of the paragraph.

9. In lines 323 to 326 we read about previous studies. In the footnote there is a reference to Taiwan on how to justify comparing studies. Isn't it better to write directly that according to previous studies conducted in Taiwan.

10. Lines 411 to 414 need to show more diligence. 

Author Response

We appreciate the effort you have taken to review our manuscript and all the suggestions you have provided. We have incorporated all the comments suggested. We hope the new changes meet their expectations, and we hope that they consider the work apt for publication in Medicina.

Comment 1: The abstract states that the study was conducted from the beginning of the pandemic. Please write clearly what time period you are referring to, the beginning and the end. The country in which the study was conducted is not stated, and the conclusions in the abstract are general in nature.

Response: Thank you for your comment. We have modified the wording of the abstract to include the dates on which our study was conducted. We have added that the study was conducted in Spain. And given more consistency to the conclusions.

Comment 2:  In the introduction it would have been appropriate to write from two sentences about the different periods of pandemic restrictions and restrictions

Response: Thank you very much for your comment. The wording of the introduction has been modified to establish the different periods of restriction in Spain.

Comment 3: In line 54 is given the date 2020, and in 59 the date 21. Does the text correspond well with each other?

Response: Thank you for your comment. The dates are different because they involve different studies. Although at the beginning of the pandemic there were abundant studies reporting on the status of HCWs, there are also later studies such as the one referenced in line 59, which dates from December 21 and talks about the prevalence of anxiety in HCWs.

Comment 4: Line 63 is not precise

Response: Thank you for your appreciation. We have modified the sentence that included line 63 to make it more coherent.

Comment 5: In line 76 to 82 there are no specific countries, and in the earlier paragraphs the names are given. It would be worthwhile to harmonize the text.

Response: Thank you for your appreciation. The wording has been modified to include the countries of the different studies to make the wording more consistent.

Comment 6: In the 94th line we read about many countries, and we have one footnote 39. So what does that mean many countries?

Response: Thank you for your comment. The wording has been modified for consistency.

Comment 7: How to explain line 130, where the questionnaire was created ad hoc?

Response: Thank you for your appreciation. We have removed the word ad hoc and reworded the sentence for consistency.

Comment 8: Item 4 is titled Discussion. Shouldn't there be added restrictions that are written about at the end of the paragraph.

Response: Thank you for your comment. The comments for the authors described at the end of the discussion have been removed.

Comment 9: 9. In lines 323 to 326 we read about previous studies. In the footnote there is a reference to Taiwan on how to justify comparing studies. Isn't it better to write directly that according to previous studies conducted in Taiwan.

Response: Thank you very much for your comment. We have modified the wording to include “according to previous studies conducted in Taiwan”. Thank you for your suggestion

Comment 10: Lines 411 to 414 need to show more diligence

Response: Thank you for your comment. This paragraph has been deleted.

Round 2

Reviewer 3 Report

Comments and Suggestions for Authors

I accept the article after correction and clarification.